# Mutual Partial Label Learning with Competitive Label Noise

**Yan Yan[1],  Yuhong Guo[1,2]**
[1]Carleton University, Ottawa, Canada     [2]CIFAR AI Chair, Amii, Canada
`yanyan@cunet.carleton.ca, yuhong.guo@carleton.ca`

## Abstract

Partial label learning (PLL) is an important weakly supervised learning problem, where each training instance is associated with a set of candidate labels that include both the true label and additional noisy labels. Most existing PLL methods assume the candidate noisy labels are randomly chosen, which hardly holds in real-world learning scenarios. In this paper, we consider a more realistic PLL scenario with competitive label noise that is more difficult to distinguish from the true label than the random label noise. We propose a novel Mutual Learning based PLL approach named ML-PLL to address this challenging problem. ML-PLL learns a prediction network based classifier and a class-prototype based classifier cooperatively through interactive mutual learning and label correction. Moreover, we use a transformation network to model the association relationships between the true label and candidate labels, and learn it together with the prediction network to match the observed candidate labels in the training data and enhance label correction. Extensive experiments are conducted on several benchmark PLL datasets, and the proposed ML-PLL approach demonstrates state-of-the-art performance for partial label learning.

## 1 Introduction

As it is costly and difficult to annotate each instance with a precise label, weakly supervised learning (WSL) has been widely studied in recent years (Zhou, 2018), which includes, but not limited to, semi-supervised learning (Van Engelen & Hoos, 2020; Ouali et al., 2020), noisy label learning (Natarajan et al., 2013; Feng et al., 2021), positive-unlabeled learning (Kiryo et al., 2017; Shu et al., 2020), and partial multi-label learning (Xie & Huang, 2018; Yan & Guo, 2021). Partial label learning (PLL) is a typical WSL problem and aims to learn a model from training samples with overcomplete labels; that is, each training sample is associated with a set of candidate labels that include both the true label and additional label noise—noisy labels. PLL has been widely applied in many real-world learning scenarios, including automatic face naming (Hüllermeier & Beringer, 2006; Zeng et al., 2013), web mining (Luo & Orabona, 2010), and multimedia content analysis (Zeng et al., 2013).

Since the ground-truth label is hidden in the candidate label set and not available to the learning algorithms, the main challenge of PLL lies in candidate label disambiguation. To address this challenge, two main label disambiguation strategies have been proposed: average-based disambiguation strategy and identification-based disambiguation strategy. Average-based disambiguation treats each candidate label equally in the model training phase and averages all the modeling outputs from each candidate label in the testing phase (Cour et al., 2011; Hüllermeier & Beringer, 2006; Zhang & Yu, 2015). Although this strategy is simple and clear, it can make the true label overwhelmed by noisy labels without sufficient differentiation, and lead to poor prediction performance. The identification-based disambiguation strategy treats the ground-truth label as a latent variable and tries to identify the true label by deriving different confidence scores for the candidate labels (Feng & An, 2018; 2019; Yao et al., 2020b; Yu & Zhang, 2016; Xu et al., 2021). The identification-based disambiguation approaches are able to achieve relatively better prediction performance than the average-based disambiguation approaches by handling the candidate labels with discrimination, but they can still suffer from the potential drawback of accumulating label identification errors and severely disrupting the subsequent model training. In addition, these existing methods are usually restricted to standard

machine learning frameworks with linear or kernel-based approaches, which have difficulties to deal with large-scale datasets.

Recently, deep learning has been widely used in addressing PLL problems (Feng et al., 2020b; Wu et al., 2022). For example, the work in (Feng et al., 2020b) proposes two PLL methods, a risk-consistent and a classifier-consistent algorithms, based on deep neural networks. The methods in (Wen et al., 2021; Lv et al., 2020) try to progressively identify the ground-truth label during training procedures by employing a self-training technique. A more recent work in (Wang et al., 2022b) tackles PLL by using contrastive representation learning with class-prototype based label disambiguation. Although it achieves satisfactory prediction performance, the contrastive learning procedure is time-consuming and resource-demanding. Moreover, all these methods have a common drawback: they automatically assume random noise in the label space; that is, they assume the candidate noisy labels are randomly sampled from a uniform generating procedure. However, during the noisy label creation process in real-world scenarios, such as crowd-sourcing, the noisy labels are typically associated with the true label and dependent on the input sample, which makes the label noise more difficult to distinguish from the ground-truth label than a random label.

In this paper, we consider a more realistic and challenging PLL learning scenario, where the noisy labels are competitive and difficult to distinguish from the true label given the input data sample. Intuitively, competitive label noise can demonstrate stronger association relationships with a true label than a random label noise, and hence are more likely to be chosen as candidate labels. For example, for online image annotation, when the object contained in an image is an "alpaca", a competitive label "camel" could have a large probability to be chosen by an annotator with limited expertise due to the similar appearances of "alpaca" and "camel", while labels such as "dog" or "duck" are less likely to be picked as part of the candidate label set due to their relatively weak association with the ground-truth "alpaca".

Motivated by the above consideration, we propose a novel and effective Mutual Partial Label Learning approach (ML-PLL) under the competitive label noise learning scenario, which learns a prediction network based classifier and a class-prototype based classifier interactively through label correction and mutual learning. Specifically, ML-PLL performs noisy label correction by integrating the outputs of both the prediction network classifier and the class-prototype based classifier, while using the corrected pseudo-labels as the targets for the prediction network training and using the output of the prediction network as target for the class-prototype based classifier training. In addition, a transformation network is proposed to model the association relationships between the true label and the noisy candidate labels, which can further enhance the classifier training with respect to ground-truth label disambiguation. Extensive experiments are conducted on several benchmark PLL datasets, while the proposed approach, ML-PLL, demonstrates state-of-the-art performance.

## 2 RELATED WORK

### 2.1 STANDARD PARTIAL LABEL LEARNING

The main challenge for addressing the PLL problem lies in how to disambiguate the candidate labels. Average-based disambiguation strategy and identification-based disambiguation strategy are two main strategies deployed in PLL. The average-based disambiguation strategy treats each candidate label equally in the model induction and averages all the modeling outputs from all the candidate labels as the final prediction. For example, the works in (Cour et al., 2011; Zhang & Yu, 2015) distinguish the averaged candidate label prediction from the non-candidate ones. The identification-based disambiguation strategy treats the ground-truth label as a latent variable and identifies the true label by deriving confidence scores for all the candidate labels (Feng & An, 2018; 2019; Yu & Zhang, 2016). For example, the works in (Zhang et al., 2016; Xu et al., 2019; Wang et al., 2022a) try to identify the true label by employing iterative label refining procedures and leveraging the topological information in the feature space. However, these methods may suffer from the cumulative errors induced in the error-prone label confidence estimation along the topological structure. In addition, a number of PLL methods propose to employ the off-the-shelf learning techniques, such as maximum likelihood, k-nearest-neighbor, maximum margin, boosting, and error-correcting output codes (ECOC), to tackle PLL problems. For the maximum likelihood technique, the likelihood of each PL training sample is defined over its candidate label set instead of its implicit ground-truth label (Liu & Dietterich, 2012). For the $k$-nearest neighbor technique, the candidate labels from neighbor

instances are aggregated to produce the final prediction on a test instance (Hüllermeier & Beringer, 2006; Gong et al., 2017; Zhang & Yu, 2015). For the maximum margin technique, the classification margin is defined over the predictive difference between the candidate labels and the non-candidate labels for each PL training sample (Nguyen & Caruana, 2008)(Yu & Zhang, 2016). For the boosting technique, the weight of each PL training instance and the confidence value of each candidate label being the truth label are refined in each boosting round (Tang & Zhang, 2017). For the ECOC technique, the binary ECOC classifiers are built based on the binary training sets filtered by the candidate labels (Zhang et al., 2017).

## 2.2 Deep Neural Network based Partial Label Learning

Standard PLL methods are usually restricted to linear based models, which have difficulty to deal with large-scale datasets. Deep learning based PLL has recently started gaining attention from the research community due to its powerful capability of training on large-scale datasets. Yao et al. (2020a) adopt deep convolutional neural networks as backbones and exploit the temporal-ensembling technique for model training. Yao et al. (2020b) propose to train two networks in a mutual learning manner, where both networks interact with each other and alleviate the error accumulation problem. Lv et al. (2020) attempt to progressively identify the true labels in a self-training manner, which can be compatibly trained with stochastic optimization. Feng et al. (2020b) assume that the probability of each candidate label being chosen is uniform and present two provably consistent methods: a risk-consistent method and a classifier-consistent method. Another work in (Wen et al., 2021) induces a family of loss functions, which generalizes the uniform generation procedures. In addition, a recent work in (Wang et al., 2022b) handles PLL by conducting class prototype-based label disambiguation following the idea of contrastive representation learning (He et al., 2020; Li et al., 2020), which achieves impressive prediction performance.

Although the abovementioned approaches produce competitive performance, most of them assume the noisy labels in the candidate label set are randomly produced, which does not hold in many real-world learning scenarios. By contrast, we tackle a more realistic and challenging PLL scenario, where the noisy candidate labels are competitive and difficult to distinguish from the true label.

## 3 Proposed Approach

In this section, we delineate the details of our proposed mutual partial label learning model (ML-PLL) in the learning scenario with competitive label noise. Given a partial label training set $D = (X, Y) = \{(\mathbf{x}_i, \mathbf{y}_i)\}_{i=1}^{n}$, where $\mathbf{x}_i \in \mathcal{X}$ denotes the $i$-th instance expressed in the input space $\mathcal{X}$, and $\mathbf{y}_i \in \{0, 1\}^L$ denotes the candidate label indicator vector associated with $\mathbf{x}_i$. The multiple 1 values in $\mathbf{y}_i$ denote either the ground-truth label or irrelevant noisy labels, which form the candidate label set $S_i \subseteq \mathcal{Y}$ for $\mathbf{x}_i$, where $\mathcal{Y} = \{1, 2, \cdots, L\}$ denotes the multi-class label space. The task of PLL is to induce a multi-class classifier from the PLL training set $D$. We assume a challenging realistic PLL scenario, where the noisy labels in the candidate label set are difficult to distinguish from the ground-truth label given the input instance.

The proposed ML-PLL model is illustrated in Figure 1, which mainly comprises the following component networks: a feature extractor $G$ that extracts features from the input instance $\mathbf{v} = G(\mathbf{x}) \in \mathbb{R}^m$; a projection network $H$ that maps the feature vector $\mathbf{v}$ into a low-dimensional embedding $\mathbf{z} = H(\mathbf{v}) \in \mathbb{R}^d$; a prediction network based classifier $F$; a class-prototype based classifier; and a transformation network $T$ that estimates the association relationships between the true label and candidate labels. The prediction network based classifier and the class-prototype based classifier are learned together through noisy label correction and interactive mutual learning, while the prediction network and the transformation network are learned together to match the observed candidate labels in the training data and enhance label disambiguation. We elaborate these components and the proposed approach in the following subsections.

### 3.1 Noisy Label Correction

The main challenge of PLL lies in that the ground-truth label coexists with the irreverent noisy labels in the candidate label set and is unknown to the learning algorithm. The key of PLL is to correct the candidate label vectors and identify the true label vectors. Given an interactive pair

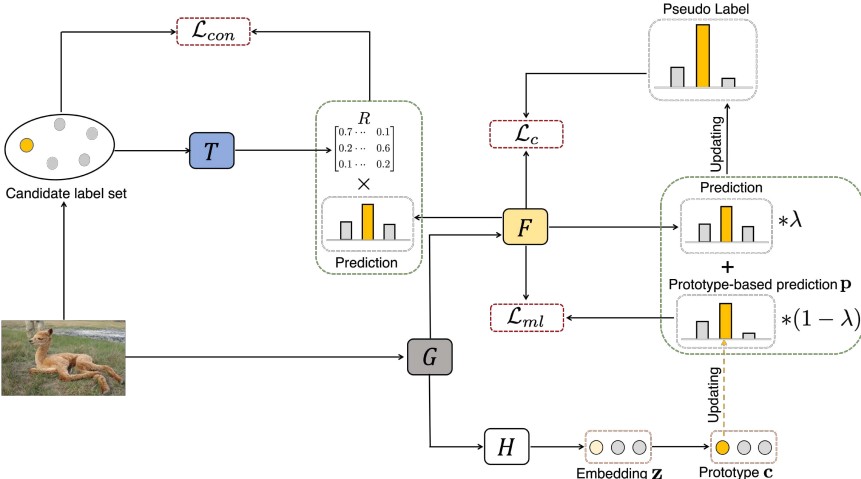

Figure 1: The proposed ML-PLL model. It learns the prediction network based classifier $F$ and the class-prototype based classifier in a mutual learning manner, while the transformation network $T$ is learned cooperatively with the prediction network to match the observed candidate labels.

of classifiers—a prediction network classifier $F$ and a class-prototype based classifier, we propose the following simple mutual label correction procedure. For any training instance $\mathbf{x}_i$, we correct its partial label indicator vector towards the true label indicator vector by exploiting the predicted pseudo-label vector. Specifically, we produce the soft pseudo-label vector $\mathbf{q}_i \in [0,1]^L$ for $\mathbf{x}_i$ by taking a weighted convex combination of the prediction network's softmax output probability vector $F(\mathbf{v}_i)$ and the class-prototype based prediction probability vector $\mathbf{p}_i$:

$$\mathbf{q}_i = \lambda F(\mathbf{v}_i) + (1 - \lambda)\mathbf{p}_i, \tag{1}$$

where $\lambda \in (0,1)$ is the combination weight parameter. Here the prediction network $F$ is built above the feature extractor $G$, such as $F(\mathbf{v}_i) = F(G(\mathbf{x}_i))$, while $\mathbf{p}_i$ is produced by the similarity-based class-prototype classifier as follows:

$$p_{ik} = \frac{\exp(\mathbf{z}_i \cdot \mathbf{c}_k/\tau)}{\sum_{k'=1}^{L} \exp(\mathbf{z}_i \cdot \mathbf{c}_{k'}/\tau)}, \tag{2}$$

where $\tau$ is a temperature parameter, $L$ is the number of class categories, $\mathbf{z}_i = H(G(\mathbf{x}_i))$ denotes the low-dimensional embedding of the features for instance $\mathbf{x}_i$, and $\mathbf{c}_k$ denotes the prototype vector for the $k$-th category, which will be elaborated in the next subsection. Instead of using Euclidean distance, we use the dot-product operator "$\cdot$" to compute the inner product similarity between the instance and the class prototypes in the embedding space for class prediction probability calculation.

To maintain an effective soft pseudo-label vector for each instance and eliminate the noise predictions outside of the candidate label set, we further restrict the soft pseudo-labels within the candidate label set and transform $\mathbf{q}_i$ into $\widehat{\mathbf{q}}_i$ with row normalization,

$$\widehat{q}_{ik} = \begin{cases} \dfrac{q_{ik}}{\sum_{j:\mathbf{y}_{ij}>0} q_{ij}} & \text{if} \quad \mathbf{y}_{ik} > 0; \\ 0 & \text{otherwise.} \end{cases} \tag{3}$$

This noisy label correction procedure can be applied on all the $n$ training instances to produce corrected soft pseudo-label vectors $\{\widehat{\mathbf{q}}_1, \cdots, \widehat{\mathbf{q}}_n\}$. With the progress of prediction network training and class-prototype update (elaborated in the next subsection), we expect each produced soft pseudo-label vector will be better aligned with the corresponding true label indicator vector, which can interactively provide a good target for the subsequent prediction network training and gradually mitigate the negative impact of the label noise during the training process.

## 3.2 BATCH-WISE MOVING AVERAGE BASED PROTOTYPE UPDATE

The effectiveness of noisy label correction relies on accurate class probability predictions, which depend on both the prediction network and the class-prototypes. Instead of training an entirely independent class-prototype based classifier, we propose to produce the class prototypes based on the output of the prediction network classifier. Specifically, we compute the class prototypes in the low-dimensional embedding space from the partial label training set $D$ in the following way:

$$\mathbf{c}_k = \frac{\sum_{\mathbf{x}_i \in D} H(G(\mathbf{x}_i)) \cdot \mathbb{1}(\widehat{\mathbf{y}}_{ik} = 1)}{\sum_{i=1}^n \mathbb{1}(\widehat{\mathbf{y}}_{ik} = 1)}, \quad \text{for } \widehat{\mathbf{y}}_{ik} = \begin{cases} 1 & \text{if } k = \arg\max_{k \in S_i} F(G(\mathbf{x}_i)) \\ 0 & \text{otherwise} \end{cases} \tag{4}$$

where $\mathbb{1}$ is an indicator function; $\widehat{\mathbf{y}}_i$ denotes predicted one-hot label vector for $\mathbf{x}_i$, which is determined by the output of the prediction network $F$ and restricted to the candidate label set $S_i$. In this way, we can automatically eliminate the prediction noise outside of the candidate labels.

With the prototype computing method above, we can update each class-prototype by recomputing it during each training iteration. However, such an update process will be computationally intensive and lead to intolerable training latency, since it requires going through the whole training set. To address this problem, we propose to perform batch-wise moving average based prototype update; that is, we update the class-prototypes for the current mini-batch by consulting the prototypes computed from previous mini-batches in a moving average manner:

$$\mathbf{c}_k = \text{Normalize}(\gamma \mathbf{c}_k + (1 - \gamma)\mathbf{c}'_k), \tag{5}$$

where $\mathbf{c}'_k$ is the mean embedding for the $k$-th class within the current mini-batch, which can be computed with Eq.(4) by limiting $D$ to the current mini-batch; $\gamma \in [0, 1]$ is a momentum coefficient; and $\text{Normalize}(\mathbf{c}) = \mathbf{c}/\|\mathbf{c}\|_2$. We expect such a batch-wise prototype update method can be computationally efficient and enhance the gradual adjustment and stability of the class prototypes.

## 3.3 PREDICTION NETWORK TRAINING WITH MUTUAL LEARNING

With the noisy label correction method above, the corrected soft pseudo-label vectors can be used as target for prediction network training. Specifically, we can learn the prediction network based probabilistic classifier on the training data $D$ by minimizing the following classification loss:

$$\mathcal{L}_c(F, G, H) = \frac{1}{n} \sum_{\mathbf{x}_i \sim D} \sum_{k=1}^L -\widehat{q}_{ik} \log(F^k(G(\mathbf{x}_i))), \tag{6}$$

where $\widehat{q}_{ik}$ denotes the $k$-th entry of the corrected soft pseudo-label vector $\widehat{\mathbf{q}}_i$ for the $i$-th instance, and $F^k(\cdot)$ denotes the $k$-th entry of the prediction network's softmax output, indicating the predicted probability of the instance belonging to the $k$-th class. We expect the prediction network based classifier $F$ can be improved by training with the corrected soft pseudo-labels.

Meanwhile, we can learn an additional class-prototype based classifier by using the output of the prediction network as prediction target and deploying the following mutual learning loss:

$$\mathcal{L}_{ml}(F, G, H) = \frac{1}{n} \sum_{\mathbf{x}_i \sim D} \sum_{k=1}^L -F^k(G(\mathbf{x}_i)) \log(p_{ik}), \tag{7}$$

where $p_{ik}$ is the predicted probability for the $i$-th instance belonging to the $k$-th class by the prototype based classifier using Eq.(2). This mutual learning loss is expected to enforce both classifiers to collaborate with each other and gradually lead to consistent predictions that can better align with the underlying true label indicator vectors.

## 3.4 TRANSFORMATION BASED LABEL ASSOCIATION LEARNING

As previously discussed, the noisy labels in the candidate label set typically have association relationships with the ground-truth label and can be competitive with the true label given the input instance in the real-world learning scenarios. In this subsection, we propose to learn the competitive association relationships between the noisy candidate labels and the true label using a transformation network $T$, aiming to enhance the label disambiguation process.

We assume the transformation network $T$ can be learned to produce a transformation matrix $R = [r_{ij}]_{L \times L}$, such that $R$ can capture the transformation relationships from a given ground-truth label to the associated noisy candidate labels and recover the candidate label set. The entry $r_{ij}$ of $R$ reflects the probability of transforming the $j$-th label to a candidate label when the $i$-th label is the ground-truth. When $r_{ij}$ is large, the $j$-th label has a strong association relationship with the $i$-th ground-truth label and hence has a larger probability to be chosen as a candidate label. Towards this goal, we propose to produce the transformation matrix $R$ by applying the transformation network $T$ on the observed candidate label matrix $Y = [\mathbf{y}_{ij}]_{n \times L}$ from the training set, such that $R = T(Y)$, since the observed candidate label matrix contains both the true labels and the associated noisy candidate labels. The transformation network $T$ can then be learned through prediction based candidate label matrix reconstruction. Specifically, we propose to reconstruct the normalized candidate label matrix by transforming the prediction outputs of the prediction network based classifier $F$ with the transformation matrix $R$. We use the symmetric Kullback-Leibler (KL) divergence to match each reconstructed candidate label distribution with its corresponding observed candidate label distribution and form the following consistency loss:

$$\mathcal{L}_{con}(F, G, H, T) = \sum_{\mathbf{x}_i \sim D} \Big( \mathrm{KL}\big(F(G(\mathbf{x}_i))R, \widetilde{Y}_i\big) + \mathrm{KL}\big(\widetilde{Y}_i, F(G(\mathbf{x}_i))R\big) \Big) \tag{8}$$

where $\widetilde{Y}_i = Y_i/\|Y_i\|_2$ is a row normalized candidate label matrix, expressing the candidate label distributions in the training set. It is however computationally intensive to compute the transformation matrix $R$ by taking the whole candidate label matrix $Y$ as input. We further propose to compute and update $R$ on each mini-batch in a moving average style, such as

$$R = \eta R + (1 - \eta)R', \tag{9}$$

where $R'$ is the transformation matrix computed from the observed partial label matrix on the current mini-batch, and $\eta \in [0, 1]$ is a momentum coefficient.

By exploiting the label association relationships between the true-label and the candidate labels, we expect the consistency loss in Eq.(8) can enhance noisy label correction and prediction network learning, thus consequently pushing the outputs of the prediction network towards the true labels.

### 3.5 Learning with ML-PLL Model

By integrating the classification loss in Eq.(6), the mutual learning loss in Eq.(7) and the consistency loss in Eq.(8) together, we get the following overall training loss for the proposed ML-PLL model,

$$\mathcal{L} = \mathcal{L}_c + \alpha \mathcal{L}_{ml} + \beta \mathcal{L}_{con}, \tag{10}$$

where $\alpha$ and $\beta$ are trade-off hyperparameters that control the relative importance of the mutual learning loss and the consistency loss, respectively. We perform training by minimizing this objective with a mini-batch based stochastic gradient descent (SGD) algorithm.

## 4 Experiment

### 4.1 Experimental Setting

**Datasets**  We conducted experiments on four widely used benchmark image datasets: Fashion-MNIST (Xiao et al., 2017), Kuzushiji-MNIST (Clanuwat et al., 2018), CIFAR-10 and CIFAR-100 (Krizhevsky et al., 2009). We transform these datasets into partial-label datasets by using the following competitive and uniform label noise generation processes. *First*, to generate competitive label noise, we train a neural network on the original clean dataset to predict the probability of each category label being the true label for a given instance, and then randomly choose the irrelevant labels with high predicted probabilities as noisy candidate labels. Specifically, assume $\hat{\mathbf{p}}_i$ is the predicted label probability vector for an instance $\mathbf{x}_i$ whose true label is class $j$. We first exclude the true label by setting $\hat{\mathbf{p}}_{ij} = 0$, and then randomly choose labels from the top-$K$ predictions in $\hat{\mathbf{p}}_i$—i.e., the $K$ irrelevant label categories with highest predicted probabilities—as the noisy candidate labels. Intuitively, these irrelevant labels with higher predicted probabilities demonstrate stronger association relationships and are more competitive with the true label. The chosen competitive noisy labels, together with the true label $j$, form the candidate label set for instance $\mathbf{x}_i$. Following this competitive

Table 1: Test accuracy (mean±std) comparison on Fashion-MNIST, Kuzushiji-MNIST and CIFAR-10 with competitive label noise at different label ambiguity levels. The best results are in bold.

| Data set | Method | Avg.#CL = 3 | Avg.#CL = 4 | Avg.#CL = 5 |
|---|---|---|---|---|
| Fashion-MNIST | ML-PLL | **90.54±0.06**% | **89.94±0.08**% | **89.37±0.09**% |
| | PiCO | 90.06±0.07% | 89.75±0.10% | 88.98±0.00% |
| | LWS | 89.31±0.05% | 88.52±0.11% | 81.07±0.05% |
| | CC | 88.69±0.13% | 88.23±0.10% | 88.05±0.36% |
| | MSE | 79.91±0.29% | 74.01±0.25% | 67.11±0.53% |
| | EXP | 89.40±0.15% | 89.15±0.10% | 88.34±0.20% |
| Kuzushiji-MNIST | ML-PLL | **95.70±0.02**% | **95.38±0.07**% | **94.87±0.07**% |
| | PiCO | 94.86±0.05% | 94.13±0.07% | 93.95±0.05% |
| | LWS | 93.04±0.06% | 92.40±0.05% | 91.21±0.06% |
| | CC | 94.58±0.32% | 94.29±0.05% | 93.10±0.06% |
| | MSE | 64.99±0.25% | 63.98±0.27% | 61.29±0.22% |
| | EXP | 93.73±0.15% | 93.54±0.17% | 92.04±0.20% |
| CIFAR-10 | ML-PLL | **84.95±0.05**% | **83.52±0.06**% | **79.25±0.09**% |
| | PiCO | 83.96±0.06% | 82.04±0.08% | 76.42±0.11% |
| | LWS | 83.15±0.21% | 79.85±0.17% | 49.65±0.25% |
| | CC | 73.12±0.44% | 69.57±0.48% | 62.11±0.43% |
| | MSE | 53.43±0.19% | 44.21±0.21% | 33.52±0.33% |
| | EXP | 73.93±0.23% | 71.24±0.22% | 51.96±0.28% |

Table 2: Test accuracy (mean±std) comparison on CIFAR-100 with competitive label noise at different label ambiguity levels. The best result in each setting is highlighted in bold.

| Data set | Method | Avg.#CL = 2 | Avg.#CL = 6 | Avg.#CL = 10 |
|---|---|---|---|---|
| CIFAR-100 | ML-PLL | **61.39±0.10**% | **57.56±0.06**% | **53.53±0.08**% |
| | PiCO | 59.86±0.10% | 54.07±0.08% | 49.22±0.05% |
| | LWS | 55.00±0.20% | 48.78±0.37% | 36.99±0.40% |
| | CC | 57.29±0.26% | 51.00±0.29% | 45.00±0.24% |
| | MSE | 39.01±0.15% | 29.76±0.11% | 18.52±0.25% |
| | EXP | 45.86±0.21% | 40.38±0.27% | 30.44±0.29% |

candidate label noise generation process, we produce partial-label datasets by choosing competitive label noise from the top-6 predictions for Fashion-MNIST, Kuzushiji-MNIST and CIFAR-10, while choosing from the top-20 predictions for CIFAR-100. The average number of candidate labels (Avg.#CL) and the characteristics of each benchmark dataset corrupted by the competitive label noise are reported in Appendix A.2. *Second*, we use the uniform generation procedure in (Lv et al., 2020) to generate uniform label noise. It generates noisy candidate labels by flipping each incorrect label with a probability $\rho$, which indicates the label ambiguity level. When encountering a special case where no irrelevant label is chosen, we randomly select one irrelevant label as the additional candidate label to ensure the label sets for all training instances are corrupted.

**Comparison Methods** We compare the proposed ML-PLL method with the following deep neural network based PLL methods: PiCO (Wang et al., 2022b) is a contrastive learning based state-of-the-art PLL method, which disambiguates the candidate labels by refining the pseudo-labels in a moving-average manner; LWS (Wen et al., 2021) is a discriminative PLL method, which considers the loss trade-off between the candidate labels and non-candidate ones; CC (Feng et al., 2020b) is a classifier-consistent method, which leverages a transition matrix to induce an empirical risk estimator; MSE and EXP (Feng et al., 2020a) are two baselines that exploit mean square error and exponential loss, respectively, as risks for inducing unbiased risk estimators. For each comparison method, we use the same backbone structure for feature extraction and adopt the same predictive model. Moreover, all the comparison methods are configured with the suggested parameters according to the source literature. For each experiment, we report the average test accuracy and standard deviation based on 5 independent runs.

**Implementation Details** We use LeNet-5 as the backbone on the Fashion-MNIST and Kuzushiji-MNIST datasets, and use 18-layer ResNet as the backbone on CIFAR-10 and CIFAR-100, for feature extraction. The projection network is a two-layer MLP that outputs 128-dimensional embeddings. The prediction network is a linear classifier. The weighted combination parameter $\lambda$ in Eq.(1), the temperature parameter $\tau$ in Eq.(2), and the momentum coefficients in Eq.(5) and Eq.(9) are set to 0.99, 1, 0.999, and 0.99 respectively. In all the experiments, we utilize a standard SGD optimizer

Table 3: Test accuracy (mean±std) comparison on Fashion-MNIST, Kuzushiji-MNIST and CIFAR-10 with uniform label noise at different label ambiguity levels. The best results are in bold.

| Data set | Method | $\rho = 0.1$ | $\rho = 0.3$ | $\rho = 0.5$ |
|---|---|---|---|---|
| Fashion-MNIST | ML-PLL | 92.75±0.05% | 92.23±0.12% | 91.86±0.08% |
| | PiCO | **93.30±0.09**% | **93.11±0.08**% | **92.78±0.03**% |
| | LWS | 91.44±0.13% | 91.85±0.14% | 90.59±0.18% |
| | CC | 92.26±0.12% | 91.75±0.04% | 90.92±0.06% |
| | MSE | 87.25±0.04% | 87.02±0.04% | 85.76±0.08% |
| | EXP | 91.02±0.06% | 89.88±0.06% | 89.21±0.11% |
| Kuzushiji-MNIST | ML-PLL | **97.63±0.07**% | **97.45±0.05**% | 96.69±0.05% |
| | PiCO | 97.58±0.06% | 97.32±0.07% | **96.95±0.03**% |
| | LWS | 96.22±0.10% | 96.15±0.24% | 95.43±0.02% |
| | CC | 96.45±0.04% | 96.16±0.02% | 95.62±0.10% |
| | MSE | 82.87±0.20% | 81.20±0.23% | 78.43±0.28% |
| | EXP | 94.98±0.03% | 94.21±0.05% | 93.56±0.11% |
| CIFAR-10 | ML-PLL | **94.64±0.11**% | 94.05±0.15% | 92.60±0.21% |
| | PiCO | 94.39±0.18% | **94.18±0.12**% | **93.58±0.11**% |
| | LWS | 90.30±0.60% | 88.99±1.43% | 86.16±0.85% |
| | CC | 82.30±0.21% | 79.08±0.07% | 74.05±0.35% |
| | MSE | 79.97±0.45% | 75.64±0.28% | 67.09±0.66% |
| | EXP | 79.23±0.10% | 75.79±0.21% | 70.34±1.32% |

Table 4: Test accuracy (mean±std) comparison on CIFAR-100 with uniform label noise at different label ambiguity levels. The best result in each setting is highlighted in bold.

| Data set | Method | $\rho = 0.01$ | $\rho = 0.05$ | $\rho = 0.1$ |
|---|---|---|---|---|
| CIFAR-100 | ML-PLL | **73.45±0.15**% | **73.06±0.13**% | **72.91±0.15**% |
| | PiCO | 73.09±0.34% | 72.74±0.30% | 69.91±0.11% |
| | LWS | 65.78±0.02% | 59.56±0.33% | 53.53±0.08% |
| | CC | 49.76±0.45% | 47.62±0.08% | 35.72±0.47% |
| | MSE | 47.45±1.50% | 45.05±1.40% | 33.27±2.81% |
| | EXP | 49.17±0.05% | 46.02±1.82% | 38.81±0.49% |

with a momentum of 0.9 and a weight decay of 1e-3 for model training. The mini-batch size, learning rate and total training epochs are set to 128, 0.01 and 400 respectively. The parameter $\alpha$ and $\beta$ are chosen from $\{0.5, 0.6, 0.7, 0.8, 0.9, 1\}$ and $\{1, 2, 3, 4, 5, 6\}$, respectively, according to the accuracy on a validation dataset (10% of the training dataset).

## 4.2 EXPERIMENTAL RESULTS WITH COMPETITIVE LABEL NOISE

We compared the proposed ML-PLL method with the five state-of-the-art comparison methods on the four datasets corrupted by competitive label noise. The comparison results on Fashion-MNIST, Kuzushiji-MNIST and CIFAR-10 with different label ambiguity levels, indicated by the average number of candidate labels, are reported in Table 1. From the table we can see that the proposed method consistently outperforms all the other comparison methods, and the performance gains yield by ML-PLL over the other methods are quite notable in many cases. For example, ML-PLL outperforms the best alternative comparison method by 1.48% and 2.83% on CIFAR-10 with Avg.#CL = 4 and 5 respectively. The experimental results on the more challenging dataset CIFAR-100, which has a large label space ($L = 100$), are reported in Table 2. From the table we can see that the proposed ML-PLL again outperforms all the other comparison methods and demonstrates remarkable performance improvement on such a complicated dataset. For example, ML-PLL improves the best comparison method, PiCO, by 1.53%, 3.49%, and 4.31% respectively across the three label ambiguity levels. Moreover, the comparison methods—e.g, PiCO, LWS, and CC—perform well with the low label ambiguity level (Avg.#CL=2), but their performance degrades dramatically with the increase of the label ambiguity level. By contrast, the proposed ML-PLL method not only consistently outperforms the other methods with notable performance gains, but also produces larger gains with higher label ambiguity levels. All these results validate the efficacy of the proposed ML-PLL approach in addressing the competitive label noise.

## 4.3 EXPERIMENTAL RESULTS WITH UNIFORM LABEL NOISE

To further validate the effectiveness of the proposed method, we also conducted experiments on the four datasets (Fashion-MNIST, Kuzushiji-MNIST, CIFAR-10 and CIFAR-100) corrupted by the

Table 5: Ablation results on Fashion-MNIST, Kuzushiji-MNIST, and CIFAR-10 with Avg.#CL = 4, and on CIFAR-100 with Avg.#CL = 6.

| Ablation variant | Fashion-MNIST (Avg.#CL = 4) | Kuzushiji-MNIST (Avg.#CL = 4) | CIFAR-10 (Avg.#CL = 4) | CIFAR-100 (Avg.#CL = 6) |
|---|---|---|---|---|
| Full Model | 89.94% | 95.38% | 83.52% | 57.56% |
| ML-PLL-w/o-ml | 86.49% | 92.18% | 80.17% | 55.27% |
| ML-PLL-w/o-con | 84.93% | 91.41% | 79.08% | 54.15% |
| ML-PLL-w/o-lc | 79.94% | 82.10% | 51.64% | 37.51% |
| CLS | 75.90% | 77.66% | 47.06% | 33.60% |

uniform label noise. The label ambiguity level in this scenario is controlled by the $\rho$ value—a larger $\rho$ probability for producing noisy candidate labels leads to larger candidate label sets. All the comparison results are reported in Table 3 and Table 4. From Table 3, we can see that although the proposed method is not specially designed for uniform label noise, it still achieves competitive performance with the best method across all cases. In Table 4, the proposed method even demonstrates substantial superiority over the other methods and produces larger performance gains with higher label ambiguity levels. These results on the challenging CIFAR-100 dataset are consistent with the ones reported in the scenario with competitive label noise—that is, the proposed ML-PLL method yields substantial and increasing performance gains when the label ambiguity level increases. All these results demonstrate the efficacy of the proposed method in handing both competitive and uniform label noise for PLL.

## 4.4 ABLATION STUDY

The objective of the proposed method contains three loss terms: the classification loss, the interactive mutual learning loss, and the consistency loss. In addition, noisy label correction is an indispensable component in the proposed mutual learning model. To validate each part's contribution, we conduct an ablation study to compare the proposed ML-PLL method with the following ablation variants: (1) ML-PLL-w/o-ml, which drops the mutual learning loss from the full model; (2) ML-PLL-w/o-con, which drops the consistency loss; (3) ML-PLL-w/o-lc, which drops the noisy label correction procedure; and (4) CLS, which only uses a standard classification loss by dropping the mutual learning loss, the consistency loss and the noisy label correction procedure. The comparison results are reported in Table 5. From the table we can see that compared with the full model, all the four variants produced inferior results, which suggests that all the components contribute to the full model in different degrees. Among the three variants of ML-PLL-w/o-ml, ML-PLL-w/o-con and ML-PLL-w/o-lc, ML-PLL-w/o-lc produces the largest performance degradation, which demonstrates the importance of the noisy label correction procedure for label disambiguation and classifier learning. Meanwhile, ML-PLL-w/o-ml and ML-PLL-w/o-con also produced consistent and very notable performance drops from the full model, which validates the contributions of both the mutual learning loss and the transformation network based consistency loss. The results suggest exploiting the association relationships between the competitive label noise and the ground-truth label can help PLL. Moreover, the large performance gap between the full model and the baseline variant CLS further validates the integrating power of the proposed method on combining all these three contributing components for PLL with competitive label noise: noisy label correction, mutual learning, and transformation network based label association exploitation.

## 5 CONCLUSION

This paper tackles a more realistic PLL scenario with competitive label noise that can be more difficult to distinguish from the ground-truth label than random noisy labels. We propose a novel mutual partial label learning approach called ML-PLL to address this challenging problem. ML-PLL learns a prediction network based classifier and a class-prototype based classifier cooperatively with label correction under a mutual learning framework. Moreover, we introduce a transformation network to model the association relationships between the ground-truth label and the candidate noisy labels, and learn it together with the prediction network through candidate label set reconstruction to enhance label correction. The comprehensive experiments on several benchmark datasets corrupted with both competitive and uniform label noise demonstrate that the proposed approach significantly outperforms the state-of-the-art PLL methods.

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

# A APPENDIX

## A.1 TRAINING ALGORITHM FOR ML-PLL

We present the mini-batch based training algorithm for ML-PLL in Algorithm 1.

---

**Algorithm 1** Training Algorithm for ML-PLL.

   **Input**: PLL training set $D$; the initialized model; trade-off hyperparameters $\alpha$ and $\beta$.

**for** s = 1: max_num_epochs **do**
    **for** t = 1: num_iterations **do**
       Sample a batch $B$ with $m$ samples from $D$.
       // Noisy label correction
       Compute the class prototype-based prediction probability vectors $\{\mathbf{p}_i\}$ via Eq.(2).
       Perform noisy label correction to get $\{\widehat{\mathbf{q}}_i\}$ via Eq.(1) and Eq.(3).
       // Batch-wise moving average based prototype update
       Compute the class prototypes via Eq.(4) on the current batch $B$.
       Update the class prototypes via Eq.(5).
       // Update of the transformation matrix
       Compute and update the transformation matrix via Eq.(9).
       // Classification loss calculation
       $\mathcal{L}_c(F,G,H) = \frac{1}{m}\sum_{\mathbf{x}_i \sim B}\sum_{k=1}^{L} -\widehat{q}_{ik}\log(F^k(G(\mathbf{x}_i)))$
       // Interactive mutual learning loss calculation
       $\mathcal{L}_{ml}(F,G,H) = \frac{1}{m}\sum_{\mathbf{x}_i \sim B}\sum_{k=1}^{L} -F^k(G(\mathbf{x}_i))\log(p_{ik})$
       // Consistency loss calculation
       $\mathcal{L}_{con}(F,G,H,T) = \sum_{\mathbf{x}_i \sim B}\left(\mathrm{KL}\big(F(G(\mathbf{x}_i))R, \widetilde{Y}_i\big) + \mathrm{KL}\big(\widetilde{Y}_i, F(G(\mathbf{x}_i))R\big)\right)$
       // Network updating
       $\mathcal{L} = \mathcal{L}_c + \alpha\mathcal{L}_{ml} + \beta\mathcal{L}_{con}$
       Update network $F$, $T$, $G$ and $H$ via stochastic gradient descent by minimizing $\mathcal{L}$.
    **end for**
  **end for**

---

## A.2 DATASET

We produce PLL datasets with different average number of candidate labels (Avg.#CL) from each benchmark dataset using the competitive label noise generation process described in subsection 4.1. The average number of candidate labels (Avg.#CL) and the characteristics of each benchmark dataset corrupted by the competitive label noise are recorded in Table 6.

## A.3 PARAMETER SENSITIVITY ANALYSIS

The trade-off parameters $\alpha$ and $\beta$ control the relative importance of the mutual learning loss and the consistency loss, respectively. We conduct experiments on CIFAR-10 at the label ambiguity level of Avg#CL = 4 to study the impacts of $\alpha$ and $\beta$ on the performance of ML-PLL. We investigate the impact of $\alpha$ by experimenting with different $\alpha$ values from $\{0.5, 0.6, 0.7, 0.8, 0.9, 1\}$ while setting $\beta = 4$, and investigate the impact of $\beta$ by experimenting with different $\beta$ values from $\{1, 2, 3, 4, 5, 6\}$ while setting $\alpha = 0.6$. The results are presented in Figure 2(a) and Figure 2(b) respectively. From Figure 2(a), we can see the best performance is achieved when $\alpha = 0.6$, while the performance difference is very small for varying $\alpha$ within $[0.5, 0.7]$. But further increasing $\alpha$ towards 1 leads to performance degradation. The results in Figure 2(b) however demonstrate larger variations with different $\beta$ values, suggesting the performance of ML-PLL is more sensitive to the $\beta$ value. When $\beta$ is very small, the performance of ML-PLL is relatively poor. With the increase of the $\beta$ value, the performance improves as the consistency loss begins to work. But when $\beta$ is too large, the performance degrades again since the consistency loss can over-dominate.

Table 6: Characteristic of the datasets corrupted by the competitive label noise generation process.

| Dataset | #Train | #Test | #Feature | Classes | Avg.#CL |
|---------|--------|-------|----------|---------|---------|
| Fashion-MNIST | 60,000 | 10,000 | 784 | 10 | 3, 4, 5 |
| Kuzushiji-MNIST | 60,000 | 10,000 | 784 | 10 | 3, 4, 5 |
| CIFAR-10 | 50,000 | 10,000 | 3,072 | 10 | 3, 4, 5 |
| CIFAR-100 | 50,000 | 10,000 | 3,072 | 100 | 2, 6, 10 |

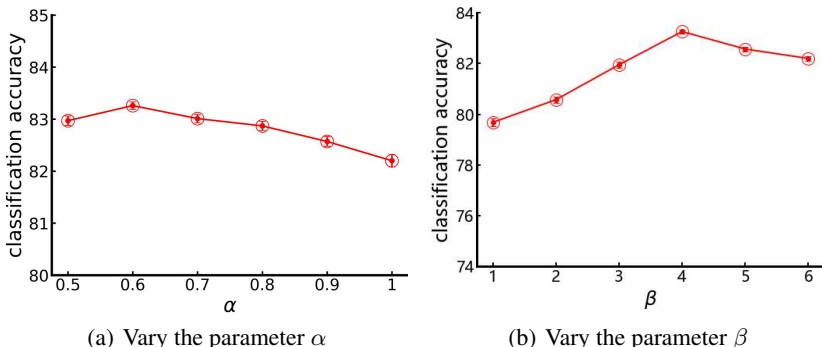

(a) Vary the parameter $\alpha$        (b) Vary the parameter $\beta$

Figure 2: Test performance with different $\alpha$ and $\beta$ values for the proposed ML-PLL on CIFAR-10.

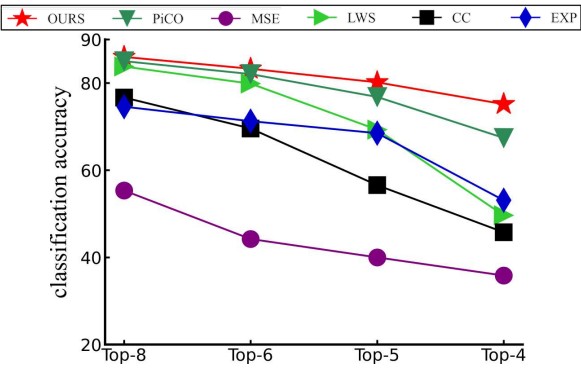

Figure 3: Impact of the competitive noise level for ML-PLL on the CIFAR-10 dataset.

## A.4 IMPACT OF COMPETITIVE NOISE LEVELS

As described in subsection 4.1, we generate the competitive label noise by randomly selecting irrelevant labels from the top-$K$ predicted categories, where $K$ controls the competitivity level of the chosen noisy candidate labels. A smaller $K$ value forces the noisy candidate labels to be chosen from the fewer irrelevant labels with the highest few predicted probabilities of being the true label, and hence indicates a higher competitive noise level. Here we conduct experiments on CIFAR-10 to investigate the impact of the competitive noise level by producing partial-label training data using different $K$ values from $\{4, 5, 6, 8\}$ but the same average number of candidate labels, Avg#CL = 4. The comparison results are presented in Figure 3. We can see that with the decrease of the $K$ value and consequently the increase of the competitive noise level, the performance of all the comparison methods degrades in general. However, the proposed ML-PLL method consistently outperforms all the other comparison methods. Moreover, the performance of ML-PLL is relatively more stable across different $K$ values, and its performance gains over the other methods become larger with the increase of the competitive noise level (the decrease of $K$), while the other methods encounter more dramatic performance degradation with smaller $K$ values. These results again validate the effectiveness of ML-PLL in addressing competitive label noise.

Table 7: Accuracy (mean±std) on CIFAR-10 with different label noise generation processes.

| Method | ML-PLL | LWS | CC | EXP | MSE |
|---|---|---|---|---|---|
| Case (1) | **94.91±0.04**% | 90.82±0.03% | 90.57±0.11% | 76.43±0.08% | 65.83±0.19% |
| Case (2) | **93.11±0.08**% | 69.42±0.07% | 69.53±0.12% | 60.47±0.14% | 40.03±0.20% |

## A.5 INFLUENCE OF NOISY LABEL GENERATION

To further investigate the impact of noisy label generation on model performance, we follow (Wen et al., 2021) to conduct additional experiments with alternative noisy label generation processes. Specifically, we generate noisy labels on CIFAR-10 with the following two commonly used label flipping matrices in the form of $[\rho_{ij}]^{L \times L}$ (Wang et al., 2022b):

$$(1) = \begin{bmatrix} 1 & 0.5 & 0 & \cdots & 0 \\ 0 & 1 & 0.5 & \cdots & 0 \\ \vdots & & \cdots & & \vdots \\ 0.5 & 0 & 0 & \cdots & 1 \end{bmatrix}, \quad (2) = \begin{bmatrix} 1 & 0.9 & 0.7 & 0.5 & 0.3 & 0.1 & 0 & \cdots & 0 \\ 0 & 1 & 0.9 & 0.7 & 0.5 & 0.3 & 0.1 & \cdots & 0 \\ \vdots & & & & \cdots & & & & \vdots \\ 0.9 & 0.7 & 0.5 & 0.3 & 0.1 & \cdots & 0 & \cdots & 1 \end{bmatrix}$$

Each entry $\rho_{ij}$ of the flipping matrix specifies the probability of the $j$-th label being selected as a candidate label when the $i$-th label is the ground-truth label. We conduct experiments with the partial label data generated using these two flipping matrices and the comparison results are reported in Table 7, where case (1) and case (2) correspond to using the partial label data generated with the flipping matrix (1) and (2), respectively. From the table, we can see that the proposed ML-PLL method outperforms all the other methods in both cases. As case (2) is more challenging than case (1), all the other comparison methods produce much inferior performance in case (2). However the proposed ML-PLL maintains similar good performance in both cases. In the simple case (1), ML-PLL outperforms the best comparison method, LWS, by 4.09%, while in the more challenging case (2), ML-PLL outperforms the best comparison method, CC, by 23.58%. These additional experimental results further validate the effectiveness of the proposed method in addressing the partial label learning problem.

