# OpenReview forum: "Mutual Partial Label Learning with Competitive Label Noise"
_ICLR.cc/2023/Conference — ICLR 2023 poster_

### Official Review · Reviewer_CWMY · 2022-10-22

**Confidence:** 4
**Correctness:** 3
**Technical Novelty And Significance:** 3
**Empirical Novelty And Significance:** 3
**Recommendation:** 6

**Clarity, Quality, Novelty And Reproducibility:**


**Clarity**

- difficult to read (in general, the presentation of the method is difficult to follow, e.g. information not yet introduced is used to explain the method and its parts).
    - One example: prototype-based classifier introduced in sec 3.2, however, sec 3.1 is all about the prototype-based classifier output and prediction probability vector p_i (based on the class-prototype classifier).

- To improve clarity (readability) it would be useful to have an example of the information flow, the steps that occur in sequence from input to output.


**Novelty**

The novelty of this paper lies above all on the idea of using competitive label noise and in part the composition of the ML-PLL architecture where the degree of novelty is slightly lower.

For example, the general structure following a mutual learning framework or the different parts that define the architecture, e.g., the "idea" behind "label correction" is well known and used in existing works on robust learning, such as, self-adaptive training a) the moving-average scheme that progressively corrects problematic labels using model predictions; b) the re-weighting scheme that dynamically puts less weights on the erroneous data.

[1] D. Tanaka, D. Ikami, T. Yamasaki, and K. Aizawa. Joint optimization framework for learning with noisy
labels. In Proceedings of the IEEE Conference on Computer Vision and Pattern Recognition, pages
5552–5560, 2018.
[2] H. Bagherinezhad, M. Horton, M. Rastegari, and A. Farhadi. Label refinery: Improving imagenet classification through label progression. arXiv preprint arXiv:1805.02641, 2018.
[3] C.-M. Teng. Correcting noisy data. In International Conference on Machine Learning, pages 239–248, 1999.
[4] C. E. Brodley and M. A. Friedl. Identifying mislabeled training data. Journal of Artificial Intelligence Research, 11:131–167, 1999.






**Strength And Weaknesses:**

**Strength**:

The idea about the new challenging PLL scenario where the noise labels in the candidate label set are difficult to distinguish from the ground-truth label given the input instance [**to be proved**].

The authors argue that in a real scenario the noisy labels are much more similar to the real one, therefore more difficult to be identified/distinguished. However, the choice about the datasets in the paper, seem to be very standard and far from the alleged "real scenario" mentioned by the authors. As the proposed approach appears to be designed to work in this setting, is it possible to empirically prove this assumption, for example on a public benchmark? A dataset which represents, more closely, the characteristics of this real scenario.


**Weaknesses**:


1. **competitive label noise** and 2. **Experiments**.

I understand the insight behind the competitive noise, but I have doubts about how this choice can change the task (simpler or harder). In general, a confounder is a variable whose presence affects the other variables studied so that the results do not reflect the actual relationship (e.g. in regression). If the confounder is chosen with very strong relationships with the GT, intuitively its presence will no longer be that of confounder.

- Would it be possible to have an experiment in which the proposed method is tested on a random selection of noise labels or the same procedure used in [5]?

To demonstrate the competitive label noise hypothesis, a comparison with some public datasets (such as those chosen in the paper) is necessary while maintaining the same split (train / test set) and choice of ground truth.

My doubt is reinforced by the comparison of the LWS method in Table 2, with the results presented in the original paper [5] on the same data (with the difference in the strategy for creating labels)

[5] Wen, H., Cui, J., Hang, H., Liu, J., Wang, Y. and Lin, Z., 2021, July. Leveraged weighted loss for partial label learning. In International Conference on Machine Learning (pp. 11091-11100). PMLR.


3. **difficult to read**


X.**Minor**:

- Equations: in general, the equations are defined as dependent on variables which then do not appear in the function definition.

One example that applies to many equations:

**Eq. 6, sec 3.3**

L_c ( F, G, H) = .... q Log(F(G(x))

q is the corrected soft pseudo-label vector.

Only going through the previous sections and Fig 1, it is possible to get that q is related to p that is related to H(G(x)). Therefore, q = Func1( F and Func2( H(G(x) ) ).


- prototype-based classifier introduced in sec 3.2, however, sec 3.1 is all about the prototype-based classifier output and prediction probability vector p_i (based on the class-prototype classifier).




**Summary Of The Paper:**

In this paper the author address the partial-label learning problem, a typical weakly supervised learning problem, where each training instance is equipped with a set of candidate labels among which only one is the true label. In this scenario, the main challenge lies in candidate label disambiguation. The authors raise the problem of the selection of noise-labels (confounders), which is usually done at random. The authors argue that in a real scenario, the noise-labels are much more similar to the real one, therefore more difficult to be distinguished. Therefore, (**contrib-1**) they propose to use noise labels that are competitive and difficult to distinguish from the true label (labels with strong relationships with GT).

The second contribution (**contrib-2**) is about the partial label learning approach that is proposed, called e Mutual Partial Label Learning approach (ML-PLL).

**ML-PLL is done by these main components**:

- They propose a noise label correction module and used the corrected pseudo-labels as the targets for the other components/networks of the proposed approach:
- F: prediction network based classifier.
- CP: Similarity based class-prototype classifier
- T: transformation network

ML-PLL learns F and CP cooperatively with label correction under the mutual learning framework. The network T is then used to estimate the relationships between the true label and candidate labels.

Experiments are performed on public datasets (Fashion-MNIST, Kuzushiji-MNIST, CIFAR-10/100) where the noise label set is hand-crafted wrt the **contrib-1**.


**Summary Of The Review:**

I'm satisfied with the response of the authors. The different points I had raised were taken into account and my doubts were clarified.

------------------------
I understand the insight behind the competitive noise, and I believe it is also a key point to evaluate the effective effectiveness of the proposed method, but I have doubts about how this choice can change the task (making it simpler or harder). In general, noise labels act by confounding, that is a causal concept, and as such, cannot be described in terms of correlations or associations. If the confounder is chosen with a particular strategy, intuitively, there is no control over whether or not its presence will be that of confounding.

To demonstrate the competitive label noise hypothesis, a comparison with some public datasets (such as those chosen in the paper) is necessary while maintaining the same split (train / test set) and choice of ground truth of one or more reference papers (e.g., 5).

[5] Wen, H., Cui, J., Hang, H., Liu, J., Wang, Y. and Lin, Z., 2021, July. Leveraged weighted loss for partial label learning. In International Conference on Machine Learning (pp. 11091-11100). PMLR.

The authors argue that in a real scenario the noisy labels are much more similar to the real one, therefore more difficult to be identified/distinguished. As the proposed approach appears to be designed to work in this setting, is it possible to empirically prove this assumption, for example on a public benchmark? Or, a dataset which represents, more closely, the characteristics of this real scenario.

---

### Official Review · Reviewer_8WLg · 2022-10-24

**Confidence:** 4
**Correctness:** 4
**Technical Novelty And Significance:** 3
**Empirical Novelty And Significance:** 3
**Recommendation:** 8

**Clarity, Quality, Novelty And Reproducibility:**

This paper is well written and the motivation of this paper is clear. The idea of considering the competitive noise labels in PLL is novel and deploy a couple classifiers to address the challenging problem in a formalized mutual learning framework, which is interesting. The optimization objective and the detailed training algorithm are provided, which seems easy to replicate the method.

**Strength And Weaknesses:**

Strengths:

1、This paper considers a more realistic PLL scenario with competitive noise labels that can be more difficult to distinguish from the ground-truth label than the random noise labels.

2、The proposed method learns a standard classifier and additional class-prototype based classifier in a formalized mutual learning framework, which is interesting.

3、The experiment results demonstrate the effectiveness of the proposed ML-PLL method in addressing competitive label noise when compared with existing PLL methods.

Weaknesses:

1、In this paper, the author learns competitive noise labels that different from the existing PLL learning scenarios. Can you give more explanation on the competitive label noise and give the difference between the ‘competitive label noise’ and ‘instance-dependent label noise’.

2、The papers propose two classifiers can be used for prediction, but it is not clear how the final prediction achieved: use the prediction network based classifier, class-prototype based classifier, or average both classifiers’ prediction as the final results.

3、How is the corrected label aligned with the ground-truth along the noise label correction? Since the ground-truth label is concealed in the candidate label set.

4、For the Figure 3 in the appendix, can you explain more about how you vary the k value to increase the competitive level.

5、In prediction network training with mutual learning, can you further explain how the two losses improve the both classifiers evaluation performance and the generalization performance.

6、The author discusses the shortcomings of the existing PLL methods that cannot be used in addressing competitive label noise, but it is unclear how can we improve the state-of-arts methods and enhance their ability to competitive label noise.


**Summary Of The Paper:**

This paper proposes a mutual learning framework called ML-PLL for partial label learning under competitive label noise. The proposed ML-PLL method disambiguates the irrelevant label noise with noise label correction and learns a couple classifiers (a prediction network based classifier and a class-prototype based classifier) simultaneously in the mutual learning framework. Experimental results validate the effectiveness of ML-PLL in addressing competitive label noise.

**Summary Of The Review:**

This paper addresses a realistic learning scenario and proposes a novel approach which makes a good contribution to partial label learning.

---

### Official Review · Reviewer_qLYR · 2022-10-24

**Confidence:** 4
**Clarity, Quality, Novelty And Reproducibility:** This paper is well written and simple…
**Correctness:** 3
**Technical Novelty And Significance:** 3
**Empirical Novelty And Significance:** 3
**Recommendation:** 8

**Strength And Weaknesses:**

Strengths:
1. The exploration of the competitive label noise in PLL is novel and the learning process is complete.
2. The paper exploits the transformation network and prediction network to match the observed candidate label distribution is interesting.
3. The experiments are relatively sufficient.

Weaknesses:
1. The motivation of the paper lacks a more detailed explanation on competitive noisy labels (shouldn't the phrases be "noisy labels" and "label noise"? and "noise" in PLL and "noise" in noisy-label learning are easily confused).
2. The instance-dependent PLL addresses the instance-dependent label noise, can you analyze the difference between the competitive noisy labels and instance-dependent noisy labels.
3. It is better to improve Figure 1, the components and the symbols in the figure are too small.
4. From the ablation study, we can see that dropping the noise label correction lead to largest performance degradation, can you explain more about it. For addressing the competitive noisy labels, the consistency loss would be more important to the full model. Moreover, from the parameter sensitivity analysis, we can also see that the consistency loss term needs large weight and important to the full model. So why dropping the noise label correction leads to more performance drop than dropping the consistency loss term.
5. What’s the difference between varying the number of averaging candidate labels and competitive level k. Does increasing the averaging candidate labels can averaging candidate labels the competitive level? Similar, does increasing the competitive level can increase the averaging candidate labels?
6. More SOTA baselins should be added, e.g., PiCo[Wang, et al., ICLR'22].

**Summary Of The Paper:**

This paper introduces a PLL approach named ML-PLL to address a more challenging setting–competitive label noise. It addresses the competitive noisy labels by following components: label correction, prediction network training with mutual learning, and transformation based label association learning, which makes couple classifiers learn and benefit from each other in a mutual learning framework thus enhancing the PLL process. Experiments validate the effectiveness of the proposed method in addressing the challenging competitive label noise.

**Summary Of The Review:**

This paper explores a challenging problem and proposes a novel mutual learning framework which makes a contribution to PLL.

---

### Decision · Program_Chairs · 2023-01-20

**Decision:**

Accept: poster

**Justification For Why Not Higher Score:**

According to my expertise and reviewing process, this paper should belong to an Accept with poster.

**Justification For Why Not Lower Score:**

According to my expertise and reviewing process, this paper should belong to an Accept with poster.

**Metareview: Summary, Strengths And Weaknesses:**

This paper focuses on a partial-label learning approach named ML-PLL, which addresses a more challenging setting called competitive label noise. To address the competitive label noise, ML-PLL has designed several key components: label correction, prediction network training with mutual learning, and transformation based label association learning. All these components make couple classifiers learn and benefit from each other in a mutual learning framework. Experiments validate the effectiveness of the proposed method in addressing the challenging competitive label noise.

The clarity and novelty are above the bar of ICLR. While the reviewers had some concerns on competitive label noise and experiments, the authors did a particularly good job in their rebuttal. Thus, all of us have agreed to accept this paper for publication! Please include the additional experimental results and further explanation in the next version.

**Note From Pc:**

if the above contains the word "oral" or "spotlight" please see: "oral" presentation means -> notable-top-5% and "spotlight" means -> notable-top-25%. As stated in our emails, we are disassociating presentation type from AC recommendations